# Predicting malaria epidemics in Burkina Faso with machine learning

**David Harvey** [1]⊛ *, **Wessel Valkenburg** [2]⊛, **Amara Amara** [2]⊛

**1** Lorentz Institute, Leiden University, Leiden, The Netherlands, **2** Terre des hommes, Lausanne, Switzerland

⊛ These authors contributed equally to this work.
* harvey@lorentz.leidenuniv.nl

**Data Availability Statement:** Our data availability is owned by the Burkina Faso government who have strictly licensed the data to Terres des hommes. The database and the algorithm developed in this study cannot be, under any circumstances shared beyond that of Terres des

## Abstract

Accurately forecasting the case rate of malaria would enable key decision makers to intervene months before the onset of any outbreak, potentially saving lives. Until now, methods that forecast malaria have involved complicated numerical simulations that model transmission through a community. Here we present the first data-driven malaria epidemic early warning system that can predict the 13-week case rate in a primary health facility in Burkina Faso. Using the extraordinarily high-fidelity data of infant consultations taken from the Integrated e-Diagnostic Approach (IeDA) system that has been rolled out throughout Burkina Faso, we train a combination of Gaussian Processes and Random Forest Regressors to estimate the weekly number of malaria cases over a 13 week period. We test our algorithm on historical epidemics and find that for our lowest threshold for an epidemic alert, our algorithm has 30% precision with > 99% recall at raising an alert. This rises to > 99% precision and 5% recall for the high alert threshold. Our two-tailed predictions have an average $1\sigma$ and $2\sigma$ precision of 5 cases and 30 cases respectively.

## Introduction

According the the World Health Organisation (WHO), in 2019 there were 229 million cases of malaria resulting in roughly 400 thousand deaths with 94% of these occurring in Africa. Of those that die, children under five are the most vulnerable, accounting for 67% of all deaths [1]. In 2019 alone, three billion dollars was contributed to programmes that aimed to control and eliminate malaria. In 2015 the WHO released a global plan that aimed to reduce cases and mortality by 90% and eliminate it from 35 countries by 2030. This ambitious goal is now to be tackled by a host of different researchers and organisations.

Recent technological advancements in computational sciences have paved the way for sophisticated analytical and numerical mathematical modelling of the transmission of malaria [2–4]. By modelling the movement of mosquitoes, weather and people in a country, these models have allowed scientists to predict the rise and fall of malaria in specific districts. These models are powerful at predicting the prevalence of malaria in areas where there is very little knowledge about the local infection rate. However, the draw backs are that they are complicated and take large amounts of computational resources to run. As a result, the continual

hommes. This relationship and trust between the charity and the Burkina Faso government must be adhered to and therefore we cannot share any of the data.The data for this study was not through a third party access but through the legal confines of Terres des hommes. All requests for data access can be addressed to the data access committee via Riccardo.lampariello@tdh.ch for researchers who meet the criteria for access to confidential data.

**Funding:** This work was in part funded by Cloudera Foundation, the Marguerite Foundation and the Delta ITP institute, and technically supported by Cloudera Foundation and Tableau Foundation.

**Competing interests:** The authors have declared that no competing interests exist.

application of these models to make real-time predictions for decision makers is non-trivial. Indeed if we are to make frequent predictions of malaria without the need to run large simulations, then we need a new way of making predictions.

## Data driven models

Machine Learning (ML) in digital health is a growing area. The rise of data-rich platforms is enabling researchers to follow a data-driven approach to modelling diseases, rather than an epidemiological approach (for a review, see [5]). This has improved all areas of health, from prevention through to post-treatment care. For example, ML is able to provide disease diagnosis at significantly earlier stages than typical methods. It was found that ensemble predictors such as Random Forests were able to predict diabetes to 93% accuracy [6]. Another study applied a gradient boosting machine to predict the risk of sepsis in patients, significantly improving on the state-of-the-art (from 49.2% to 67.7%) [7]. Furthermore researchers have used genetic algorithms to predict heart disease in its early stages [8] and lastly, a combination of principal component analysis and neural networks have been used to diagnose diabetic retinopath in patients [9].

ML has also aided the analysis of patient data improving treatment. For example, support vector machines and Random Forests are able to localise brain tumours in magnetic resonance images [10]. Finally, machine learning is aiding post-care treatment with algorithms now able to predict which patients are at a high risk of readmission, a significant cost to healthcare systems [11], and algorithms that are able to monitor public opinion on how healthcare systems manage certain crises. For example, [12] used social media to scrape public opinions of how people reacted to COVID-19. It is clear from the breadth of examples that machine learning can advance healthcare in a multitude of ways.

Malaria specific data-driven models are rare given the lack of structured data-sets. [13] carried out a study looking at a variety of different machine-learning techniques to predict Malaria cases in healthcare centres in the district of Visakhapatnam, India. Using 6 years worth of data they found that Gradient Boosting was a good predictor of cases. However, this was limited to just one small region of India, with no inter-healthcare variance.

Despite the opportunity that machine-learning presents, risks exist. Biases in poorly selected training and tests samples (for example through insufficient representation of a given class in the sample [9]), can result in biased decision making once the algorithm has been implemented [14]. It is clear that going forward, if machine-learning is to be used to improve the delivery of healthcare, it must be done so with caution and transparency [15].

In this paper we attempt to directly model the case rate of malaria in Burkina Faso over the last three years. In this way we can provide in-situ predictions of future case rates immediately, without the need to run complicated numerical simulations of malaria transmission in the country. Moreover, by directly modelling the case rate we encompass not only the transmission but other factors, including social, to provide extremely localised predictions.

## IeDA

The Integrated e-Diagnostic Approach (IeDA, https://www.tdh.ch/en/ieda) is an implementation of the Integrated Management of Childhood Illness, IMCI [16], on Android-based tablets. The application allows primary healthcare workers to conduct efficient and informed consultations for children younger than 5 years. The primary aim of the IMCI protocol is to combat child mortality, which in the case of Burkina Faso fluctuates around 8% [17]. The IeDA program started in 2010 in 39 primary healthcare facilities (PHC) in Tougan, Burkina Faso. Since then (at time of writing), it has risen to cover more than 67% of the entire country, taking

around 200, 000 consultations each month. This has already dramatically improved the delivery of healthcare to Burkina Faso, particularly in rural areas. As a result of the IeDA project there has been a reduction in antibiotic usage by $\sim 6 - 15\%$ [18] and the potential savings of millions of Swiss Francs.

For those PHCs in the program, the tablet is used in 92% of children's consultations, such that we have a good coverage of this population. The database is fully pseudonymised to retain the privacy of the patients and contains over 9 million consultations. Throughout this study we use the confirmed diagnosis of malaria as the target feature. This may have its own biases, which we discuss later. The objective of this paper is two fold:

1. Predict the trajectory of the case numbers of malaria for a specific primary healthcare facility;

2. Train and validate an early warning system for the onset of an epidemic up to three months in advance.

## Epidemic detection

The definition of an epidemic is defined as the point at which the instantaneous case rate rises above the five-year mean for that period of year plus two standard deviations. The disadvantage of this is that it is an instantaneous definition, such that a region does not know it is in an epidemic until it arrives. Early warning of the onset of an epidemic will allow local and federal governments to prepare and react before the onset of an emergency. It is the aim of this work to develop an early warning system for malarial epidemics in Burkina Faso up to three months in advance.

## Materials and methods

### Patient and public involvement

We utilise the IeDA database that consists of consultations of infants less than an age of five years. All data is fully anonymised. Patients agree to be part of the database through a verbal agreement. Data has been provided by the Burkina Faso government under strict licensing uniquely to Terres des hommes for the purpose of improving healthcare delivery throughout Burkina Faso.

### Time series

The base framework of the algorithm is to fit a library of models to some observed data that will then predict the subsequent time period required. For example, we will use models which span 26 weeks. This will allow us to fit to 13 weeks of observed data and then for the model to predict 13 weeks ahead in time.

This library of models will be agnostic in time and geographical location, that is, to make a single prediction we will compare a single observed time series to the entire library of time series that we have and find the best fitting ones.

We therefore first manipulate the historical data vectors into series of time windows of 26 weeks. With $\sim 1000$ PHCs and $\sim 2.5$ years of data we have in total $\sim 100, 000$ separate time series that we can use to formulate a library of models.

### Learning each time series: Gaussian Processes

A naive method would be to use the raw data as our library of models, comparing historical pasts directly with the current data. However, the historical past suffers from large variance

due to not just random noise but also noise generated from social factors such as strikes and routine monthly cycles of medicine distribution. If we are to generate a library of reliable models that look beyond weekly variations and extract the seasonal cycles of malaria and how these correlate with other observables then we must use a more sophisticated model. To do this we choose a Gaussian Process.

A Gaussian Process (GP from now on), is defined as a collection of finite random variables that are Gaussian distributed. Each variable can be fully described by their mean, $m$ and their co-variance, $k$, where we assume a exponential, sine squared kernel,

$$k(x_i, x_j) = \exp\left(\frac{2 \sin^2(\pi d(x_i, x_j)/p)}{l^2}\right),$$ (1)

where $d$ is the distance between observed values, $x_i$ and $x_j$, $p$ is the periodicity and $l$ is the length scale, both free parameters to be fit. Indeed we test these free parameters and find that the impact on these data-sets is small. We choose a periodicity of $p = 52$, corresponding to the annual cycle of 52 weeks, and the length scale, $l = 1$.

Following these choices we then fit the GP to our data. During the fitting we define how well we expect the GP to do at describing the data, i.e. to avoid over-fitting to each time series we give it some buffer. This acts to choose functions that are smoother since they are statistically a better fit. Not only this, however but co-variances *between* observables, builds a model that jointly fits each observable equally. As such the advantages of a GP is that it takes in to account correlations not only in time but between observables.

We use the freely available Gaussian Process Regressor (GPR) package from scikit-learn [19] to model our time series. We jointly fit each GPR to each chosen time-dependent feature, assuming some level of noise. This noise must be tuned empirically to strike a balance between fitting to signal in the data and over-fitting to noise. The top panel of Fig 1 shows the impact of a varying noise parameter, $\alpha$, on the modelled time series of absolute malarial cases. We see that in the low $\alpha$ limit the GPR finds models that fit the data *exactly*, greatly over-fitting, whereas a very high noise value results in the GPR fitting only the mean of the time series.

To tune $\alpha$ we calculate the log marginal likelihood of each model given the data-set and find the noise value that returns the maximum. The log marginal likelihood of a Gaussian Process takes in to account how well the data fits the model, and how complex the model is. From Bayes Theorem, the marginal likelihood, or evidence, of the target $\mathbf{y}$ given the input data $X$, is the product of likelihood of the target values given the function $f$, $p(\mathbf{y}|\mathbf{f}, X)$, and the prior over $f$, $p(\mathbf{f}|X)$, integrated over the parameters of the function $f$, i.e.

$$p(\mathbf{y}|X) = \int p(\mathbf{y}|\mathbf{f}, X)p(\mathbf{f}|X)d\mathbf{f}.$$ (2)

For a Gaussian Process, the prior is a Gaussian with a mean of zero and co-variance, $k$, $\mathcal{N}(0, k)$ and the likelihood of the function given the data is also a Gaussian with a mean of the input function $f$ and the input variance, $\alpha$. [19] show by carrying out the integral in Eq (2), the log marginal likelihood is,

$$\log(p(\mathbf{y}|X)) = -\frac{1}{2}(\mathbf{y} - f)^{\mathrm{T}}(k + \alpha^2 I)^{-1}(\mathbf{y} - f) - \frac{1}{2}\log|k + \alpha^2 I| - \frac{n}{2}\log 2\pi,$$ (3)

where $I$ is an identity matrix and $n$ is the number of observations. We see that the first term of the marginal likelihood is how well the data is fit by the Gaussian Process, the second term penalises complexity in the algorithm. To find the best fitting $\alpha$ parameter we carry out a grid based search, testing different levels of noise and finding the mean marginal log-likelihood.

(A)

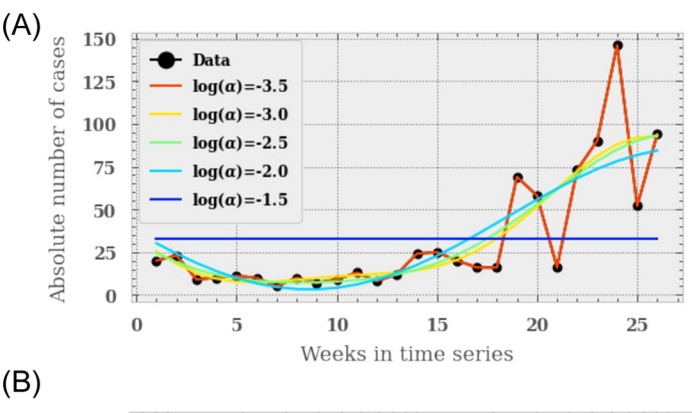

(B)

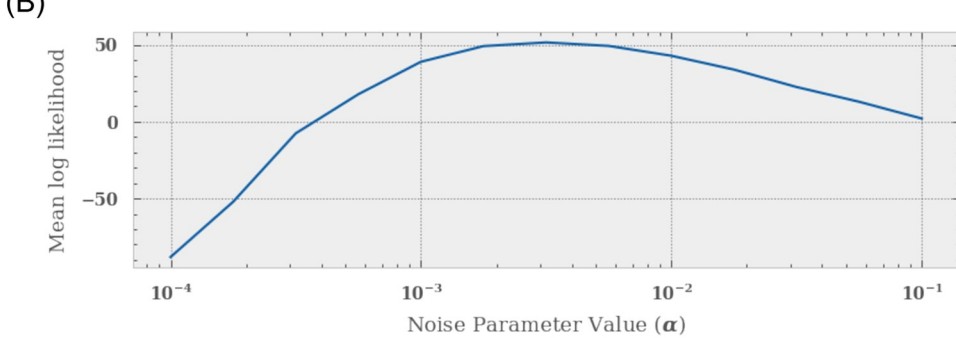

**Fig 1.** *Top*: How the fit of the Gaussian Processor Regressor to the time series data depends on the noise value $\alpha$. Assuming a sine, exponential squared kernel, we see that for small $\alpha$ values the regressor over-fits the malaria (top), fitting every point in each curve. Additionally, at high values of *alpha*, the regressor tends to the mean of each time series. *Bottom*: We optimise the noise value through a simple grid-based search for the best fitting value of $\alpha$. For each time-series we derive a log-likelihood of the model given the data. We then find the *alpha* parameter that has the best mean log-likelihood over the entire data-set. We find that the log(*alpha*) = −2.75 returns the highest mean likelihood.

The bottom panel of Fig 1 shows the result of this search. We find that a value of $\log(\alpha) = -2.75$ best fits the data.

## Model selection

Now with a library of models based on historical data, we want to be able predict the case rate of malaria for a given PHC. To do this, we take the observed data from the previous 13 weeks and compare this to the first half of every model in our library. We construct a joint likelihood of each model and its component features and find the best fitting one. From this we can take the second half of these models to predict the following 13 weeks.

We construct the total log-likelihood for a given observed data-set, $\mathcal{L}_\mathrm{T}$, as the linear addition of each observable's log-likelihood, $\mathcal{L}_i$, i.e.

$$\mathcal{L}_\mathrm{T} = \sum_{i=0}^{\mathrm{nObs}} \mathcal{L}_i, \tag{4}$$

where we assume either a Gaussian log-likelihood,

$$\mathcal{L}_\mathrm{G} = -\frac{1}{2}\chi^2 = -\frac{1}{2}\frac{(D-M)^2}{\sigma_M^2}, \tag{5}$$

where $D$ and $M$ are the data and model respectively and $\sigma$ is the total error or Poisson,

$$\mathcal{L}_\mathrm{p} = -D + D\ln M - \ln(M!), \tag{6}$$

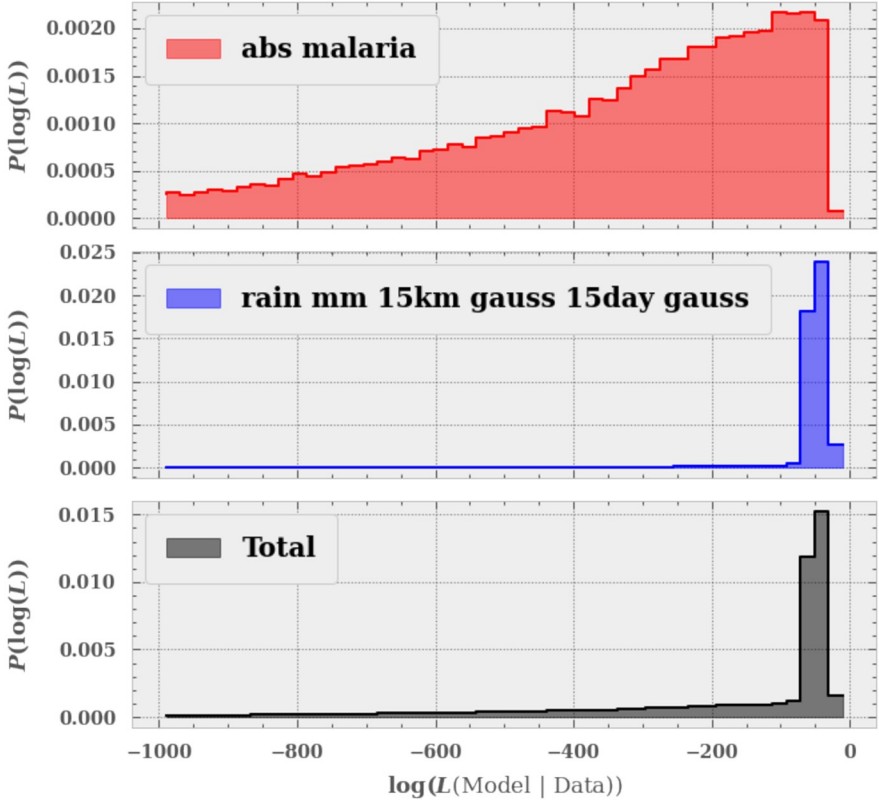

**Fig 2. An example of how we select our models.** We compare the observed data to each GP and generate a likelihood according to the type of data (whether it be Poisson or Gaussian). In this case the top panel showing the absolute number of cases of malaria is Poisson distributed, whereas the rain-fall is Gaussian. We then combined these likelihoods in to a final likelihood and select only those models that have an overall good fit.

where this is the log-likelihood of observing D events for a Poisson distribution with a mean of *M* events. Fig 2 shows an example of the likelihood function. Each panel shows the distribution of the likelihood of a model given a single observed data set for the absolute case numbers of malaria (top) and the rain fall (middle). The bottom panel shows the total likelihood given the two observables. We see that the final distribution of likelihoods has a small number of models that fit well and the rest that do not.

We then select all models that lie within some $\Delta\mathcal{L}_\mathrm{T}$ threshold, $\lambda$, relative to the best fitting model's log-likelihood. We define this as the $\Delta\chi^2$ for the given number of degrees of freedom, where the $\Delta\chi^2$ is given by,

$$\Delta\chi^2 = -2\Delta\mathcal{L}_\mathrm{T}. \tag{7}$$

Despite the assumption of Gaussian statistics, PHCs have extremely irregular time series that result in models that are equally irregular. This nuance with the data lead us to two decisions: (1) we need to tune the value of $\Delta\chi^2$ which selects models that fit a time-series at hand, and (2) from the selected models we need to derive what the predictions are at various levels of confidence.

This is not a trivial operation since a model may have a very low $\Delta\chi^2$ (fitting the time-series very well), but still give a prediction which is extremely rare when compared to other models with a same low $\Delta\chi^2$. In order to make a prediction we therefore carry out an iterative process: We first choose an initial threshold, find the prediction and estimate the error bars. The

second iteration will increase the threshold slightly and make another prediction. The iterative approach continues until the error bars are stable to within 5%. This ensures that the estimated error bars are stable and not sensitive to the addition of a couple of models, moreover, the adaptive threshold allows the algorithm to account for time series with different noise properties. How this initial threshold and the sub-sample selection is carried out depends on the desired confidence region (see section).

## Features

It is known that observables such as rain correlate well with forthcoming epidemics of malaria [20, 21]. It is therefore clear that introducing these in to our model can improve the selection process. We can introduce observables trivially by including them in the model fitting of the GPR and then the likelihood through Eq (4). The feature set to adopt is potentially very large since the available data is broad. However, not all features will contribute information equally. We set-out an initial framework to understand the sensitivity of the algorithm to these different features, these include: the absolute number of confirmed cases of malaria; the absolute number of consultations; absolute number of tests required; the confirmed number of cases of malaria within a 30km Gaussian smoothed region (not including the current centre); confirmed number of malaria cases within 100km (not including the current centre); rain-fall and surface water. We then carry out a series of tests to determine whether or not each feature adds predictive power. We find that only rain-fall improves the precision of the algorithm. We therefore select our primary base of absolute malaria cases and rain-fall as our feature-base.

### Estimating the amplitude: Random Forest Regressor

In addition to the standard observables we can also introduce meta-observables to use during the fitting. For example, a key variable in making the prediction is understanding the overall normalisation of the time series (i.e. the mean case number). This way we can remove many models that rise too early, or do not rise at all.

In order to introduce this meta-variable in to our feature set we set-up a classic machine-learning problem: using 13 weeks prior information can we estimate the 26 week mean case rate. To do this we initialise a random-forest regressor (RFR) [22]. We set up feature base where we calculate the mean and standard deviation of each observable over the 13 week time period. The target feature is the 26 week mean of the absolute malaria case. Then using an Extra Trees Random Forest Regressor from the scikit-learn package we train a random forest.

We first split the historical data in to a test and training set. We take all data before the 1st May 2020 as the training set and all data after as the test set. For the training data we compile our feature set of the 13 week mean and variance for all available features (a total of 14 features) and the target feature as the 26 week mean of the absolute malaria numbers. We then carry out a simple grid-based search we determine the optimum number of trees and minimal split sample, which turn out to be 1000 and 2 respectively.

We show the results of our test sample in Fig 3. The left hand panel shows the true versus the estimated normalisation for the test set with the solid blue line showing the 1–1 correlation and the red markers the mean and variance in the estimate. We see that there is a mild bias in our estimate, often over-estimating the true normalisation. We find that the RFR has an error of ∼30%. The right hand panel shows the estimated importance of each feature when estimating the normalisation, with the caption giving the legend. We find that the absolute number of confirmed malaria cases, absolute number of consultations and the absolute number of tests needed (and their standard deviations) are strong predictors, with the others adding only a small amount of information.

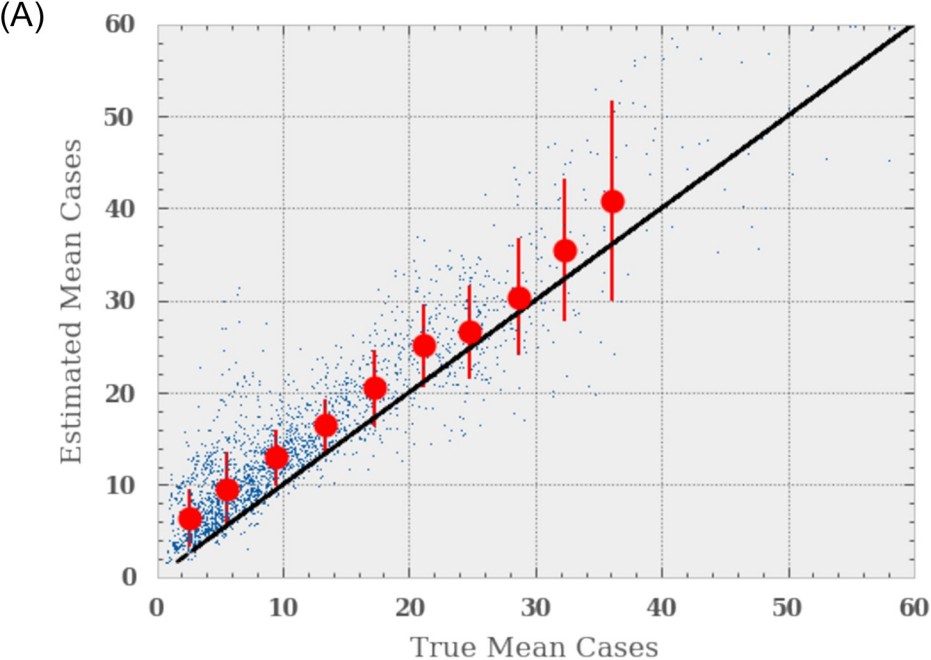

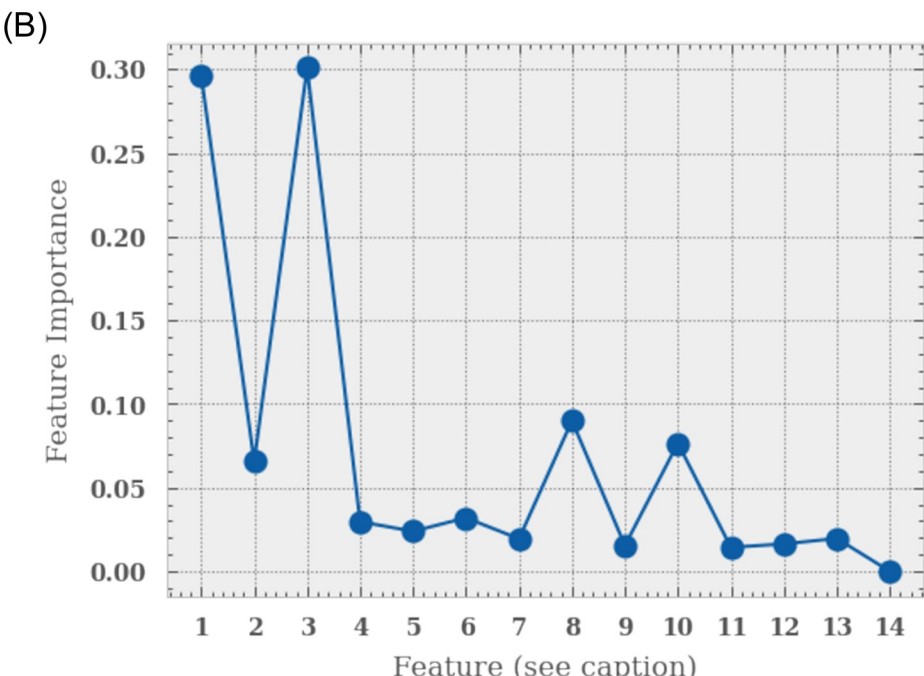

**Fig 3. The performance of the Random Forest Regressor at estimating the mean malarial case rate over a 26 week period using data from the first 13 weeks.** We set up a feature base and train the regressor. The top panel shows the estimated mean case rate as a function of the true case rate for the test set. The red markers show the mean and variance in binned true case rates. We find that the estimated case rate has a slight bias. The bottom panels shows feature importance from the Random Forest. The legend is as follows, the first seven as the mean 13 value for (1): absolute number of malaria cases, (2):the absolute number of consultations, (3): the absolute number of tests needed, (4): the Gaussian smooth case rate of malaria over a 30km region, (5): the Gaussian smooth case rate of malaria over a 100km region, (6):the rain-fall in mm over a 15km Gaussian smoothed area and 15day time period, (7): the amount of surface-water, and the variance in each feature over the same 13 week window.

We find that introducing this normalisation constraint improves our model by a factor of two in its precision.

## Calibrating the errors: One and two-tailed uncertainties

The threshold ($\Delta\chi^2$) in which we cut the number of models we include in our estimate is somewhat arbitrary. As such we must calibrate it in order to ensure that the $\Delta\chi^2$ corresponds directly to the confidence region we state.

The algorithm that we have developed will have two key objectives. The first is a warning system; whereby we alert the user that the number of malaria cases are expected to rise above the threshold for an epidemic in the coming three months. This constitutes *a lower bound* and hence a one-tailed probability distribution. The second objective is a forecast such that the algorithm sets out a lower and upper bound between which it believes the case rates of the malaria will follow; this constitutes a two tailed probability distribution. As such we calibrate the error to four distinct uncertainties: the two tailed $1\sigma$ and $2\sigma$ and the one tailed $1\sigma$ and $2\sigma$ upper and lower bound.

Since the lower the threshold the smaller the uncertainties, we search for the lowest possible threshold that remains consistent with the desired confidence region. We use a Monte Carlo Markov Chain to find this lowest threshold.

## Results

Following the description of the algorithm we now present its performance. In order to calibrate and test our algorithm we split our archival data in to a test and training sample. We split the entire data-set in time from the 1st May 2020 (such that it agrees with the same split for the RFR). This ensures that there is $\sim 1:10$ split in the amount of test to training data (such that the expected uncertainty in the prediction will be close to using the full data-set for the final algorithm). For each test we take a time series of 26 weeks and split it into two, the first 13 weeks which is the data we use to make our prediction, and the following 13 weeks as our ground truth that we can compare to.

We split our tests in to three distinct scenarios: the rising, (where the case numbers rise over the time series); falling, (where case numbers fall through out the time series); and flat, (where case numbers remain relatively stable throughout the time series). This enables us to examine exactly how the algorithm performs in different malarial scenarios. Moreover, it ensures that we do not over-fit and bias towards one specific scenario that may be more common in the data-set (since we take from the 1st May 2020 there will be one scenario that is more common that others). We also test the six different estimators: the 68% and 95% two-tailed precision, and the 68% and 95% one-tailed upper and lower precision. This ensures that the estimated confidence intervals are consistent with all scenarios. The reader can find the results of the accuracy tests in Appendix A in S1 Appendix.

Fig 4 shows the results from the one-tailed precision tests. The left hand panel shows precision of the lower bounds for the three different scenarios and the 1&2$\sigma$ confidence regions (i.e. 68% and 95%) as a function of the week predicted. The right hand panel shows the same except for the upper bound.

We find that the lower bounds are most precise in the falling scenario and the least in the rising scenario. The 1$\sigma$ lower bound has a 25 case precision in the 13th week in a rising case, improving to a 5 case precision in the 13th week for the falling scenario. We find that the 2$\sigma$ bounds are $\sim$ 60% looser than the 1$\sigma$ bounds, rising from 8 case precision in the 13th week for the falling scenario to 40 case precision in the rising scenario for the same week.

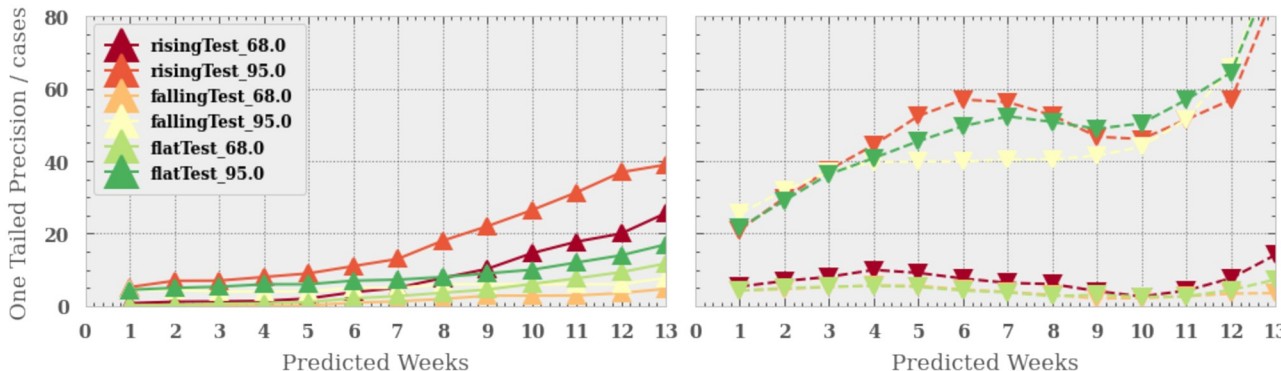

**Fig 4. The one-tailed precision for the algorithm as a function of predicted week for the three scenarios and two confidence limits (68% and 95%).**
The left hand Fig shows the precision for the lower confidence limit and the right hand Fig the precision for the upper confidence limit.

The upper bounds are more consistent between scenarios, particularly in the early weeks of the prediction. However, in the rising scenario the upper bounds become more precise. This is because the aggregation of many exponential models acts to smooth out the rise and therefore the estimated bounds are no longer exponential. As such the bounds do not match precisely the shape of the exponential rise in cases. As a result, in the rising case scenario, towards later time, the true line tends towards the upper bound and away from the lower bound, increasing the precision in the upper bound and reducing it in lower bound. We find that the $1\sigma$ upper bound for the all scenarios are less than 15 cases even at the 13th week, whereas the $2\sigma$ interval is significantly worse at 80 cases for the falling scenario.

Fig 5 shows the precision of the two-tailed confidence region for the three scenarios as a function of the week predicted. We find that the algorithm performs best in the falling case scenario for both confidence intervals and the worst for the rising case. This is unsurprising since when cases fall they do so slowly and linearly, where as when cases rise they do so exponentially and therefore much more difficult to predict. We find that in the best case scenario that we have a two-tailed precision at the $1\sigma$ ($2\sigma$) of 5 (26) cases and 15 (52) in the worse case scenario.

## Precision and recall

We now estimate the precision and recall of the algorithm at estimating the occurrence of epidemics. It is important to distinguish between 'claimed epidemics' (the algorithm raises an

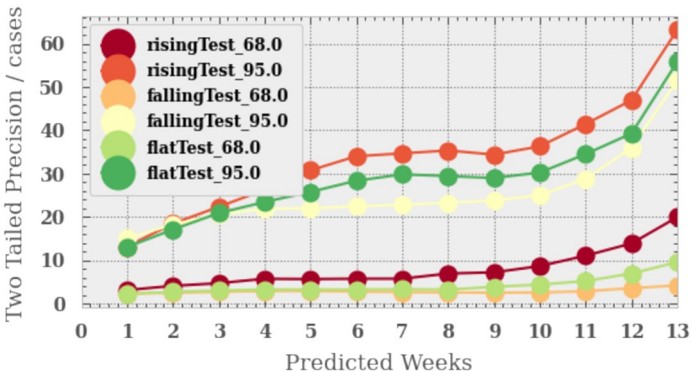

**Fig 5. The two-tailed precision for the algorithm as a function of predicted week for the three scenarios and two confidence intervals (68% and 95%).**

alert, regardless of what eventually happens), and 'actual epidemics'. Precision is defined as the fraction of claimed epidemics that are actual epidemics, and recall is the fraction of actual epidemics which were claimed. Formally, they are defined as

$$\text{Precision} = \frac{tp}{tp + fp} \qquad \text{Recall} = \frac{tp}{tp + fn}, \qquad (8)$$

where $tp$ = the number of true positives, $fp$ = the number of false positives and $fn$ = the number of false negatives. In order to do this we must create a test set that has known epidemics in them. The World Health Organisation's definition of an epidemics is when the case numbers rise above the five year average plus two standard deviations. Unfortunately the database only has three years worth of historical information. We therefore create a threshold around the three year mean plus two standard deviations. We create a test set of 131 situations where epidemics have occurred and a test set of 300 scenarios where no epidemic has occurred, again with a minimum date of the 1st May 2020 to ensure not future data exists in the training set.

In order to test the precision and recall of our algorithm we must now make clear what and when our algorithm defines an epidemic. Given the confidence intervals of the prediction, there will be different percentages of confidence in different situations. To avoid confusion, let us now use percentiles rather than confidence intervals. The $x$th percentile is the value below which $x$% of the predictions fall. So if the 5th percentile curve crosses the WHO definition of an epidemic, this means that 95% of our models predict an epidemic. Conversely, if only the 95th percentile exceeds the WHO definition, this means that 5% of the models predict an epidemic. It should be straightforward to see now, how choosing the percentile at which we trigger an epidemics directly translates into a degree of confidence at which the epidemic is going to happen.

In brief: when the 5th percentile equals the WHO epidemic definition, we are 95% confident that the epidemic will happen, and vice versa. As a direct consequence, using a low percentile as the trigger guarantees a high precision: the alert is only raised when we are highly confident that an epidemic will happen, and as a consequence only true epidemics are claimed, but also many true epidemics are missed. There is a high number of false negatives. Using a high percentile for the trigger, guarantees a high recall: all true epidemics are claimed, together with a high number of false-positives and few false-negatives.

The top panel of Fig 6 shows how we define our epidemic alert system. We show the probability distribution for the number of cases for a given week and PHC. Should the WHO definition of an epidemic cross a certain percentile, the system will trigger an alert. The bottom panel shows the corresponding precision and recall as a function of these trigger percentiles (that must exceed the threshold before an epidemic is claimed). For example the left-most points show the recall (red, star) and precision (blue, balls) when the 95th percentile goes above the threshold. The blue line gives the precision and the red the recall. We find that when we claim an epidemic as soon as the 95th percentile exceeds the threshold, we return 100% of the epidemics, however with only 30% precision. I.e 70% of the time we claim an epidemic when in-fact there isn't one. When using higher percentiles, we find that our precision improves to 100% with a recall of 4.5%. Table 1 gives the results of these tests. We note here that the total number of positives in the test set was 131 and negatives in the test set 300 therefore we have a limit of 0.76%, therefore should our algorithm get all correct then this results in a > 99% (since 100% is clearly impossible). Finally we also would like to note that these percentiles are how we classify as our tier grading system in the alert system (see section).

(A)

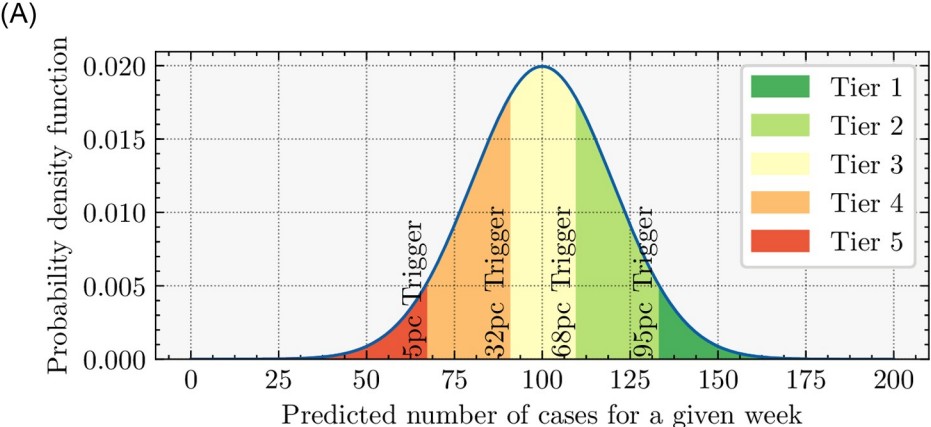

(B)

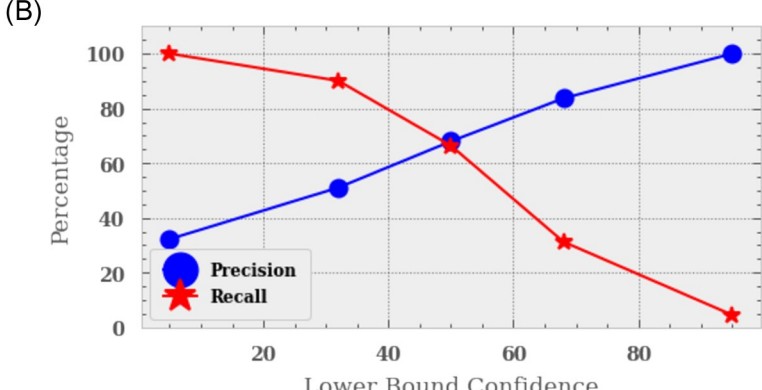

**Fig 6.** *Top*: How we trigger an alert for an epidemic. Here we show the predicted number of cases for a given week (of thirteen) and PHC. Each colour shows a different tier of alert and the associated percentile trigger. Should the WHO definition of an epidemic cross one of those triggers, an alert is made, with the associated precision and recall. *Bottom*: The precision and recall of the algorithm corresponding to these trigger levels. The precision (blue) is the fraction of true positives the algorithm makes from the total claimed positives and the recall (red) is the fraction of true positives from the total number of positives (or epidemics) there are. Here we show these as a function of the proportion of models predicting a case rate above the threshold. For those bounds that include more models that predict the case rate to go above the threshold we get a higher precision, however a lower recall.

## Implementation

Following our error calibration and precision and recall testing, we now look at how this algorithm will be implemented. We choose to have a tier system with our alerts since it is not clear whether an optimal precision or recall is best suited. As such we have generated a five-tier

**Table 1. The precision (fraction of detected epidemics are indeed epidemics) and recall (the rate at which we claim an epidemics in the case of an epidemic) as a function of the percentile (i.e. the proportion of models predicting a case rate *below* the threshold).** For example, when we raise an alert when the 95th percentile goes above the threshold, we detect all epidemics, however, 68% of the time the claimed epidemics do not result in an epidemic; simply put we claim epidemics all the time.

| Percentile Trigger | Precision (%age) | Recall (%age) |
|---|---|---|
| 5th | > 99 | 4.580 |
| 32nd | 83.673 | 31.298 |
| 50th | 67.969 | 66.412 |
| 68th | 51.082 | 90.076 |
| 95th | 32.266 | > 99 |

system based on Fig 6. Each tier is defined by how many of these limits cross the threshold for an epidemic at any point during the 13 weeks. There are upper percentiles: 5%, 32% 68%, 95% which correspond to five tiers (with the first being when no percentile crosses the threshold).

1. Tier 1: No Alert: All percentiles are below the threshold for an epidemic throughout the 13 week period. There is a negligible chance ($< 1\%$) that cases will rise above the threshold during the next 13 weeks.

2. Tier 2: Low Alert: The 95th percentile (only), crosses the threshold for an epidemic. The algorithm suggests that there is a small chance that cases may rise above the epidemic threshold. Based on the empiric performance of the model (Table 1), there remains a 70% chance no epidemic occurs.

3. Tier 3: Medium Alert: The 95th and 68th percentiles cross the threshold for an epidemic. There is a 50/50 chance of an epidemic at some point during the 13 weeks.

4. Tier 4: High Alert: The 95th, 68th and 32nd percentiles cross the epidemic threshold: There is a 84% chance of an epidemic.

5. Tier 5: Very High Alert: All four percentiles at some point during the 13 weeks are above the threshold for an epidemic: There is a greater than 99% chance of an epidemic.

In addition to the alert system, this algorithm also provides a two-tailed prediction. This will allow the user to understand how long the epidemic will last, and the magnitude of the epidemic. As such the algorithm will provide two-tailed confidence regions. Fig 9 in S2 Appendix in the appendices shows a set of ten examples. In each case we show the preceding 13 weeks of data to which we fit our models. We then show the following 13 week prediction. In yellow (orange) we should the $1\sigma$ ($2\sigma$) two-tailed confidence region. The green (red) lines with the arrows signify the upper and lower bounds at the $1\sigma$ ($2\sigma$) confidence. We stress here that these lower and upper bounds are independent of one another unlike the confidence region.

## Discussion

In this study we have shown that we can estimate the future cases of malaria. This algorithm has been implemented in to the Burkina Faso governmental database to aid decision making at the district level. However, there remains limitations:

1. All rates of malaria are based on data from the confirmed diagnosis of malaria in the IeDA database, that consists only of infants less than five years old. Extrapolation to the full population requires an understanding of how these two demographics correlate. This correlation could be age dependent, geography dependent and time dependent. This needs to be looked in to, however is beyond the scope of this work.

2. The base idea behind the algorithm is that it uses historical data to predict the trajectory of malaria cases going forward. Therefore the predictions are dependent on the quality of the data. Should the data be biased for one reason or another then these maybe propagated in to the predictions. For example, should a district enact a intervention policy, whereby they distribute a large amount of medicine to a region then this will alter the trajectory of the malarial cases in that area. Currently this algorithm will use this case and a non-intervened case as an equal possibility in its prediction and hence they will be degenerate with one another. It is therefore vital to include extra, external data that can break these degeneracies. These can include, for example, information regarding intervention policies, local natural and social events that prevent people from going to hospital (and hence recording malaria cases), biases within the consultation that may result in certain demographics having a

better recorded malaria prevalence. These biases should be explored within the current data-set to see how the predictions, precision and accuracy changes with each one, however, is beyond the scope of this work.

3. Implementation of the machine-learning algorithms is a highly debated one, with even the simplest of data representation (risk maps) being not completely effective [23]. Translating error bars and risk analysis to interventions is something that must be carried out beyond the machine-learning domain, and exists as a continuing discussion between data-scientists, epidemiologists and key decision makers. As such this work represents the foundation in a long tree of decisions that must be made in order to have true impact.

## Conclusion

We have presented a combined Gaussian Process and Random Forest Regressor that can predict the case rate of malaria to within 5 and 30 cases at the $1\sigma$ and $2\sigma$ confidence level.

Using a combination of historical data from the Integrated e-Diagnostic Approach (IeDA) database of consultations of infants less than five years old in Burkina Faso and external rain data, we construct a library of Gaussian Processes that we use to fit to observed data to make a 13 week prediction. We calibrate these errors to ensure they are accurate to one and two standard deviations. We find that although only rain and malaria data aid our algorithm in select models that predict well the trajectory of malaria cases, features such as the absolute number of consultations and the variance in this statistic are also a good predictor of the expected daily rate of malaria cases.

We create a three scenario test set: the first, a scenario where cases rise up (akin to an epidemic); the second, a scenario where case numbers remain flat and thirdly a scenario where case numbers fall. We find that our algorithm is least sensitive in the scenario where case numbers rise and the most precise when the case numbers fall. This is due to the difficulty in modelling the fast, exponential rise in the case rate, whereas the fall is often linear and easier to predict.

We calibrate and test a five tier epidemic alert system based on the lower bound threshold of the algorithm. The lower-bound threshold alert is based on the situation whereby we make an alert when the lower limit of our predicted case rate (at some given confidence interval) goes above the threshold for an epidemic (the five-year mean plus two standard deviations). For example, the 95% confidence lower-bound is the limit whereby we are 95% confident the case rate will be above this prediction. Should this limit be above the threshold for an epidemic, we will be extremely confident an epidemic is going to occur in the forthcoming 13 weeks. We test and validate this alert system. We find that the precision of epidemic prediction of this system for the 95th, 68th, 32nd and 5th percentile is 32%, 51%, 83% and > 99% respectively. However, the recall rate for the same lower-limits are > 99%, 90%, 66% and 5% respectively.

We address potential biases in the algorithm, highlighting the need to introduce further features that will break potential degeneracies that exist due to social factors, such as strikes and malaria intervention programmes. Moreover, further work highlighting potential demographic biases within the system need to be addressed.

Finally we note that the algorithm is limited. The current IeDA database only contains consultations of infants less than five years old and therefore predictions must be scaled to the overall population. How this is scaled is not a trivial problem to solve. Moreover the data is only three years old and therefore has covered a limited number of malaria seasons, and it currently does not cover the entirety of Burkina Faso. As such, in its current form it may not be as good a predictor as current epidemiological methods (for example directly simulating the

transmission of the disease). Such a comparison would be interesting going forward. Despite these limitations, this work represents the first efforts to develop a data-driven predictor of malaria in sub-Saharan Africa.

## Supporting information

**S1 Appendix. Accuracy tests.**
(PDF)

**S2 Appendix. Examples.**
(PDF)

## Acknowledgments

The authors thank Seydou Toguiyeni, Aziza Merzouki, Maroussia Roelens, Iveth Gonzalez, Antoine Geissbuhler (University of Geneva), Beatriz Galatas (WHO), the IeDA team of Terre des hommes in Burkina Faso and the Ministry of Health from Burkina Faso for their input and discussions.

## Author Contributions

**Conceptualization:** David Harvey, Wessel Valkenburg, Amara Amara.

**Data curation:** David Harvey, Wessel Valkenburg.

**Formal analysis:** David Harvey, Wessel Valkenburg.

**Funding acquisition:** Amara Amara.

**Investigation:** David Harvey, Wessel Valkenburg.

**Methodology:** David Harvey, Wessel Valkenburg.

**Project administration:** David Harvey, Wessel Valkenburg, Amara Amara.

**Resources:** Amara Amara.

**Software:** David Harvey, Amara Amara.

**Supervision:** Amara Amara.

**Validation:** David Harvey, Wessel Valkenburg.

**Visualization:** David Harvey.

**Writing – original draft:** David Harvey.

**Writing – review & editing:** David Harvey, Wessel Valkenburg, Amara Amara.

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
