## [Decision Letter · Decision Letter 0]

13 Apr 2021

PONE-D-21-09535

Predicting Malaria Epidemics in Burkina Faso with Machine Learning

PLOS ONE

Dear Dr. Harvey,

Thank you for submitting your manuscript to PLOS ONE. After careful consideration, we feel that it has merit but does not fully meet PLOS ONE’s publication criteria as it currently stands. Therefore, we invite you to submit a revised version of the manuscript that addresses the points raised during the review process.

Based on the comments from the reviewers and my own observation, I recommend major revisions for the paper; 

We look forward to receiving your revised manuscript.

Kind regards,

Thippa Reddy Gadekallu

Academic Editor

PLOS ONE

Journal Requirements:

In your Data Availability statement, you have not specified where the minimal data set underlying the results described in your manuscript can be found. PLOS defines a study's minimal data set as the underlying data used to reach the conclusions drawn in the manuscript and any additional data required to replicate the reported study findings in their entirety. All PLOS journals require that the minimal data set be made fully available. For more information about our data policy, please see http://journals.plos.org/plosone/s/data-availability.

Thank you for stating the following in the Acknowledgments Section of your manuscript:

The authors thank Seydou Toguiyeni, Aziza Merzouki, Maroussia Roelens, Iveth Gonzalez, Antoine Geissbuhler (University

of Geneva), Beatriz Galatas (WHO), the IeDA team of Terre des hommes in Burkina Faso and the Ministry of Health from Burkina

Faso for their input and discussions. This work was in part funded by Cloudera Foundation, the Marguerite Foundation and the

Delta ITP institute, and technically supported by Cloudera Foundation and Tableau Foundation.

The authors have declared that no competing interests exist.

We note that Figure C 11 in your submission contain map images which may be copyrighted. All PLOS content is published under the Creative Commons Attribution License (CC BY 4.0), which means that the manuscript, images, and Supporting Information files will be freely available online, and any third party is permitted to access, download, copy, distribute, and use these materials in any way, even commercially, with proper attribution. For these reasons, we cannot publish previously copyrighted maps or satellite images created using proprietary data, such as Google software (Google Maps, Street View, and Earth). For more information, see our copyright guidelines: http://journals.plos.org/plosone/s/licenses-and-copyright.

4a, You may seek permission from the original copyright holder of Figure C 11 to publish the content specifically under the CC BY 4.0 license. 

4b, If you are unable to obtain permission from the original copyright holder to publish these figures under the CC BY 4.0 license or if the copyright holder’s requirements are incompatible with the CC BY 4.0 license, please either i) remove the figure or ii) supply a replacement figure that complies with the CC BY 4.0 license. Please check copyright information on all replacement figures and update the figure caption with source information. If applicable, please specify in the figure caption text when a figure is similar but not identical to the original image and is therefore for illustrative purposes only.

Reviewers' comments:

Reviewer's Responses to Questions

**Comments to the Author**

1. Is the manuscript technically sound, and do the data support the conclusions?

Reviewer #1: Yes

Reviewer #2: Yes

2. Has the statistical analysis been performed appropriately and rigorously? 

Reviewer #1: Yes

Reviewer #2: Yes

3. Have the authors made all data underlying the findings in their manuscript fully available?

Reviewer #1: Yes

Reviewer #2: Yes

4. Is the manuscript presented in an intelligible fashion and written in standard English?

Reviewer #1: No

Reviewer #2: No

5. Review Comments to the Author

Reviewer #1: The authors have presented a a combined early warning system and malaria predictor that can predict the 13 week trajectory of malaria cases in an primary health facility in Burkina Faso. This paper is suitable for publication but it needs minor revision.

Below are my comments:

• The contributions of the authors are not clear. They have mentioned in first contribution.

• Several paragraphs contain trivial information and should be dropped.

• Each section should have a summary table. If contents are too much, then add summary tables for the subsection.

• Each section should present new information and perspective to enlighten the readers.

• Paper contributions are presented without pitching problems in recent studies. Add one paragraph before it to highlight issues in recent studies and at the end how this paper overcome those shortcomings.

• Improve the presentation and resolution of Fig. 3,4,5,6. It’s a very informative figure.

• writing is good, need to check the typo errors.

• Paper is well-formatted, plz check the formatting of the reference

• I found some English mistakes please check them.

• There is lot of relevant literature missing. Please cite the below articles but not limited to:

a) Bojja, Giridhar Reddy, Martinson Ofori, Jun Liu, and Loknath Sai Ambati. "Early public outlook on the coronavirus disease (COVID-19): A social media study." (2020).

b)Reddy, G. Thippa, M. Praveen Kumar Reddy, Kuruva Lakshmanna, Dharmendra Singh Rajput, Rajesh Kaluri, and Gautam Srivastava. "Hybrid genetic algorithm and a fuzzy logic classifier for heart disease diagnosis." Evolutionary Intelligence 13, no. 2 (2020): 185-196.

c) Rehman, Zaka Ur, M. Sultan Zia, Giridhar Reddy Bojja, Muhammad Yaqub, Feng Jinchao, and Kaleem Arshid. "Texture based localization of a brain tumor from MR-images by using a machine learning approach." Medical hypotheses 141 (2020): 109705.

d) Bojja, Reddy, and Omar El-Gayar. "Predicting Hospital Readmissions of Diabetic patients-A Machine Learning Approach." (2019).

Reviewer #2: 1. The authors have to proofread the article carefully. There are many grammatical errors in the paper.

2. There are several long sentences in the paper. Try to use simple and short sentences.

3. The abstract has to be rephrased.

4. WHat are the limitations of the existing works that motivated the current research?

5. List out the main contributions of the current work.

6. The related work can be summarized as a table.

7. Some of the recent and relevant works on disease prediction and machine learning such as the following can be discussed in the paper: "Variance ranking attributes selection techniques for binary classification problem in imbalance data, An adaptive multi-layer botnet detection technique using machine learning classifiers, Early detection of diabetic retinopathy using PCA-firefly based deep learning model".

8. Compare the current work with recent state-of-the-art.

9. Discuss about the limitations of the current work in conclusion.

6. PLOS authors have the option to publish the peer review history of their article (what does this mean?). If published, this will include your full peer review and any attached files.

Reviewer #1: No

Reviewer #2: No

---

## [Author Response · Author response to Decision Letter 0]

21 May 2021

Dear Editor

We would like to thank yourself and the reviewers for the very detailed and useful comments. Below is our response to each point (firstly the editor's and then the reviewers'). We have highlighted the amended text in the manuscript in bold.

We hope with these adjustments the paper can be accepted for publication.

Kind Regards

David Harvey, Wessel Valkenberg and Amara Amara

Editor

1. We have now altered the format of the manuscript to align with the journal requirements. 

2. Our data availability is owned by the Burkina Faso government and strictly licensed to Terres des homes with no exceptions. The database and the algorithm developed in this study cannot be, under any circumstances shared beyond that of Terres des homes. This relationship and trust between the charity and the Burkina Faso government must be adhered to and therefore we cannot share any of the data. 

However, as confirmed by the referees review, we have provided sufficient evidence throughout that we have carried out a rigorous statistical analysis.

3. We apologise for the funding confusion. We have removed the statement from the manuscript and would like our statement replaced with.

“This work was in part funded by Cloudera Foundation, the Marguerite Foundation and the Delta ITP institute, and technically supported by Cloudera Foundation and Tableau Foundation.” 

4. We have now removed this figure from the manuscript.

Reviewer #1

• The contributions of the authors are not clear. They have mentioned in first contribution.

We thank the referee for pointing this out. We have specifically added a section at the end of the paper. 

• Several paragraphs contain trivial information and should be dropped.

We thank the referee for suggesting reducing the size of the paper. We have now significantly reduced its length without loss of clarity.

• Each section should have a summary table. If contents are too much, then add summary tables for the subsection.

We are not entirely sure we understand what the reviewer is suggesting here. 

• Each section should present new information and perspective to enlighten the readers.

We have now reduced the size of the paper such that each section hopefully meets this criterion.

• Paper contributions are presented without pitching problems in recent studies. Add one paragraph before it to highlight issues in recent studies and at the end how this paper overcome those shortcomings.

We thank the referee for suggesting we reorganize the introduction. We have now re-written it.

• Improve the presentation and resolution of Fig. 3,4,5,6. It’s a very informative figure.

We thank the referee, we have improved the figures now.

• writing is good, need to check the typo errors.

We have proof read the manuscript again. We apologise if a few have slipped through the net.

• Paper is well-formatted, plz check the formatting of the reference

We thank the referee for this and have altered the formatting to be in line with the journal.

• I found some English mistakes please check them.

We have proof read the manuscript again. We apologise if a few have slipped through.

• There is lot of relevant literature missing. Please cite the below articles but not limited to:

We thank the referee for suggesting these articles. We have reviewed the introduction and added more references.

Reviewer #2: 

1. The authors have to proofread the article carefully. There are many grammatical errors in the paper.

We have proof read the manuscript again. We apologise if we have missed a few.

2. There are several long sentences in the paper. Try to use simple and short sentences.

We thank the referee for pointing this out. We have tried to reduce the paper and simplify it without loss of clarity.

3. The abstract has to be rephrased.

We have now rephrased the abstract and hope it is suitable.

4. What are the limitations of the existing works that motivated the current research?

We now motivate the work more clearly at the end of “Data driven models”

5. List out the main contributions of the current work.

We thank the referee for pointing this out. We have specifically added a section at the end of the paper. 

6. The related work can be summarized as a table.

Does the referee mean the author contributions? If so we have added a section now and hope that means the criteria.

7. Some of the recent and relevant works on disease prediction and machine learning such as the following can be discussed in the paper.

We thank the referee for their comments. We have now added the relevant references to the introduction.

8. Compare the current work with recent state-of-the-art.

9. Discuss about the limitations of the current work in conclusion.

We thank the referee for these important comments. Indeed including the limitations in the conclusions is vital and have therefore added this. However, comparison to state-of-the-art is difficult, because there isn’t one currently. However, we have noted a comparison with simulations.

---

## [Decision Letter · Decision Letter 1]

2 Jun 2021

Predicting Malaria Epidemics in Burkina Faso with Machine Learning

PONE-D-21-09535R1

Dear Dr. Harvey,

We’re pleased to inform you that your manuscript has been judged scientifically suitable for publication and will be formally accepted for publication once it meets all outstanding technical requirements.

Kind regards,

Thippa Reddy Gadekallu

Academic Editor

PLOS ONE

Additional Editor Comments (optional):

Reviewers' comments:

Reviewer's Responses to Questions

**Comments to the Author**

1. If the authors have adequately addressed your comments raised in a previous round of review and you feel that this manuscript is now acceptable for publication, you may indicate that here to bypass the “Comments to the Author” section, enter your conflict of interest statement in the “Confidential to Editor” section, and submit your "Accept" recommendation.

Reviewer #1: All comments have been addressed

Reviewer #2: All comments have been addressed

2. Is the manuscript technically sound, and do the data support the conclusions?

Reviewer #1: Yes

Reviewer #2: Yes

3. Has the statistical analysis been performed appropriately and rigorously? 

Reviewer #1: Yes

Reviewer #2: Yes

4. Have the authors made all data underlying the findings in their manuscript fully available?

Reviewer #1: Yes

Reviewer #2: Yes

5. Is the manuscript presented in an intelligible fashion and written in standard English?

Reviewer #1: Yes

Reviewer #2: Yes

6. Review Comments to the Author

Reviewer #1: Authors have done a great job in addressed the points raised by the reviewers. Now the manuscript looks more precise and sound.

Reviewer #2: The authors have addressed all the suggestions and comments I recommend accepting the paper in its present form.

7. PLOS authors have the option to publish the peer review history of their article (what does this mean?). If published, this will include your full peer review and any attached files.

Reviewer #1: No

Reviewer #2: No

---

## [Editor Report · Acceptance letter]

9 Jun 2021

PONE-D-21-09535R1 

Predicting Malaria Epidemics in Burkina Faso with Machine Learning 

Dear Dr. Harvey:

I'm pleased to inform you that your manuscript has been deemed suitable for publication in PLOS ONE. Congratulations! Your manuscript is now with our production department. 

Kind regards, 

on behalf of

Dr. Thippa Reddy Gadekallu 

Academic Editor

PLOS ONE